

# Predictive value of three Inflammation-based Glasgow Prognostic Scores for major cardiovascular adverse events in patients with acute myocardial infarction during hospitalization: a retrospective study

Houyong Zhu[1,2,*], Zhaodong Li[3,*], Xiaoqun Xu[3], Xiaojiang Fang[2], Tielong Chen[2] and Jinyu Huang[4]

[1] Department of Cardiology, Hangzhou Hospital of Traditional Chinese Medicine (Dingqiao District), Hangzhou, Zhejiang, China
[2] Department of Cardiology, Hangzhou Hospital of Traditional Chinese Medicine (Wulin District), Hangzhou, Zhejiang, China
[3] Department of Clinical Laboratory, Hangzhou Red Cross Hospital, Hangzhou, Zhejiang, China
[4] Department of Cardiology, The Affiliated Hangzhou First People's Hospital, Zhejiang University School of Medicine, Hangzhou, Zhejiang, China
* These authors contributed equally to this work.

Corresponding authors
Tielong Chen, ctlppp@foxmail.com
Jinyu Huang, hjyuo@foxmail.com

## ABSTRACT

**Aim:** Inflammation-based Glasgow Prognostic Scores (GPS) have been reported to predict the prognosis of patients with acute ST-elevation myocardial infarction (STEMI) undergoing primary percutaneous coronary intervention (PPCI). The goal of this study was to investigate whether three kinds of GPSs can effectively predict major cardiovascular adverse events (MACEs) in STEMI or non-ST-segment elevation myocardial infarction (NSTEMI) patients undergoing PPCI, elective PCI (EPCI) or conservative drug therapy during hospitalization.

**Methods:** In this retrospective cohort study, patients with acute myocardial infarction (AMI) were divided into 0, 1 or 2 score according to the GPSs. Logistic regression and receiver operating characteristic (ROC) curve analysis were performed to assess the predictive value of GPSs for MACE and all-cause mortality during hospitalization. Three kinds of GPSs, Inflammation-based Glasgow Prognostic Score (GPS), modified GPS (MGPS) and high-sensitivity CRP-modified GPS (HS-MGPS) and Global Registry of Acute Coronary Events (GRACE) score were applied in this study.

**Results:** A total of 188 patients were enrolled. The ROC curve with MACE showed that the AUC of GPS (0.820 (95% confidence interval (CI) [0.754–0.885]), $P < 0.001$) was larger than that of MGPS (0.789 (95% CI [0.715–0.863]), $P < 0.001$), HS-MGPS (0.787 (95% CI [0.717–0.856]), $P < 0.001$) and GRACE score (0.743 (95% CI [0.672–0.814]), $P < 0.001$). The ROC curve with all-cause mortality showed that the AUC of GPS (0.696 (95% CI [0.561–0.831]), $P = 0.005$) was similar to the HS-MGPS (0.698 (95% CI [0.569–0.826]), $P = 0.005$) and higher than the MGPS (0.668 (95% CI [0.525–0.812]), $P = 0.016$), but lower than the GRACE score (0.812 (95% CI [0.734–0.889]), $P < 0.001$). Multivariate logistic regression analysis showed

that the GPS was an independent risk factor for the incidence of MACE during hospitalization. Compared with the odds ratio (OR) value for a GPS of 0, the OR for a GPS of 1 was 7.173 (95% CI [2.425–21.216]), $P < 0.001$), and that for a GPS of 2 was 18.636 (95% CI [5.813–59.746]), $P < 0.001$), but not an independent risk factor for all-cause mortality ($P = 0.302$). GRACE score was an independent risk factor for MACE (1.019 (95% CI [1.004–1.035]), $P = 0.015$) and all-cause mortality (1.040 (95% CI [1.017–1.064]), $P = 0.001$). In the subgroups classified according to the type of AMI, the presence of disease interference GPSs and the type of PCI, the ability of GPS to predict the occurrence of MACE seemed to be greater than that of MGPS and HS-MGPS.

**Conclusion:** The GPS has a good predictive value for the occurrence of MACE during hospitalization in patients with AMI, regardless of STEMI or NSTEMI, the choice of PCI mode and the presence or absence of diseases that interfere with GPS. However, GPS is less predictive of all-cause mortality during hospitalization than GRACE score, which may be due to the interference of patients with other diseases.

## INTRODUCTION

Despite the development of percutaneous coronary intervention (PCI), coronary artery bypass grafts and standardized revascularization strategies, acute myocardial infarction (AMI) is still one of the leading causes of mortality worldwide. In Europe, unselected patients with AMI, especially ST-segment elevation myocardial infarction (STEMI), still have a hospital mortality rate of 4% to 12% and an annual mortality rate of approximately 10% (*Ibanez et al., 2018*). In the United States, STEMI patients have a hospital mortality rate of 5–6% and an annual mortality rate of 7–18% (*O'Gara et al., 2013*). However, in China, the mortality rate of AMI during hospitalization is higher than that in Europe and the United States (*Chen et al., 2017*). The efficient prediction and prevention of major adverse cardiovascular events (MACEs) are considered effective measures to avoid death outcomes (*Shang et al., 2019*). Although age, heart failure, renal function and the number of coronary artery lesions can be used to predict the adverse outcomes of AMI, these indicators cannot clearly and intuitively evaluate the prognosis of AMI patients. The inflammation-based Glasgow Prognostic Score (GPS) makes use of the changes in albumin and hypersensitive C-reactive protein (H-CRP) to concisely evaluate the prognosis of cancer patients. In recent years, a retrospective study (*Çınar et al., 2019*) showed that the ratio of H-CRP to albumin can predict the adverse outcome of patients with STEMI and prospective cohort studies by *Jia et al. (2018)* also indicated that the GPS also has a good predictive ability for the prognosis of STEMI patients undergoing primary PCI (PPCI). However, whether the GPS can effectively predict MACEs in STEMI or non-ST-segment elevation myocardial infarction (NSTEMI)

patients undergoing PPCI, elective PCI (EPCI) or conservative drug therapy has not been reported.

Therefore, for the first time, we conducted a clinical trial to assess the predictive power of the GPS combined with two other modified GPSs for adverse events in patients with AMI during hospitalization.

## MATERIALS AND METHODS

### Subjects

This was a single-center, retrospective cohort study designed to assess the predictive value of three GPSs for adverse events during hospitalization in patients with AMI undergoing PPCI, EPCI or conservative drug therapy. Patients were recruited from the Department of Cardiovascular Disease of Hangzhou Hospital of Traditional Chinese Medicine from 1 January 2016 to July 24, 2019. This study belonged to retrospective cohort study, and the exemption of informed consent had been approved by the ethics committee of Hangzhou hospital of traditional Chinese medicine (Ethical Application Ref: 2019KY028). This study protocol strictly complied with the requirements of the Helsinki Declaration of the World Medical Association and the international ethics guide for human biomedical research of the Council for International Organizations of Medical Sciences (CIOMS).

### Patient selection

According to the results of the preliminary test, the optimal threshold was obtained through the Youden index, and the sensitivity of the best expectation was 0.862, while the specificity of the best expectation was 0.714. Finally, the estimation of the sample size of the diagnostic test was completed according to the expected sensitivity and specificity (*Obuchowski & Zhou, 2002*; *Li & Fine, 2004*), and its sample size was at least 184, which was calculated by MedSci Sample Size tools (http://m.medsci.cn/sci/sample_size_diagnosisSamplerate.do).

According to the third global definition of myocardial infarction (*Thygesen et al., 2012*), a total of 188 patients met the inclusion criteria during the retrospective retrieval period. The exclusion criteria were as follows: (1) lack of data on H-CRP and serum albumin; (2) critical patients discharged automatically without MACEs; and (3) patients who lack or refuse to sign informed consent during hospitalization.

### Data collection

The baseline data included sex, age, diagnosis, Killip classification, MACE, PCI type, acute infection, autoimmune diseases, tumors, nephrotic syndrome, uremia, cirrhosis, hypertension, diabetes, blood pressure, ejection fraction (EF) and biochemical indicators including hemoglobin, platelet, type B natriuretic peptide (BNP), D-dimer, alanine aminotransferase (ALT), low density lipoprotein (LDL), creatinine, albumin, H-CRP, troponin I (TNI), creatine kinase (CK) and creatine kinase MB (CK-MB). All biochemical blood tests were performed in the laboratory for clinical purposes. All biochemical indicators were selected as the first biochemical results after admission. The GPS, modified

GPS (MGPS), high-sensitivity CRP-modified GPS (HS-MGPS) and Global Registry of Acute Coronary Events (GRACE) score (*Granger et al., 2003*) were calculated. The score of GRACE was completed by reference to *Granger et al. (2003)*, *Fox et al. (2006)*, *Eagle et al. (2004)*. In short, Killip classification, systolic blood pressure, heart rate, age, creatinine and whether there was pre hospital cardiac arrest, ST segment down shift and myocardial enzyme elevation to form GRACE score. H-CRP and albumin were detected by latex particle-enhanced turbidimetric immunoassay and bromcresol green method, respectively. EF was detected by the M-type method and Simpson method. In addition, because the GPSs were composed of H-CRP and albumin, patients with acute infections, autoimmune diseases, tumors, nephrotic syndrome, uremia or liver cirrhosis were classified into the high tendency interference GPS group (HTI-GPS), and the rest were classified into the low tendency interference GPS group (LTI-GPS).

## Definition of GPS

The definition of GPS was as follows: patients with an increased H-CRP level (>10 mg/L) and a low albumin level (<35 g/L) are designated as having a GPS of 2. The presence of one abnormality associated with either the H-CRP level or albumin level is designated as a GPS of 1. If both indicators are normal, a GPS of 0 is designated.

The MGPS strengthens the position of H-CRP in the score, which is defined as follows: patients with an increased H-CRP level (>10 mg/L) and low albumin level (<35 g/L) are designated as having an MGPS of 2. Patients with abnormal H-CRP levels are designated as having an MGPS of 1. As long as the H-CRP is not abnormal, it is designated as an MGPS of 0.

The HS-MGPS further strengthens the position of H-CRP in the score, which is defined as follows: patients with an increased H-CRP level (>3 mg/L) and low albumin level (<35 g/L) are designated as having an HS-MGPS of 2. Patients with abnormal H-CRP levels are designated as having an HS-MGPS of 1. As long as the H-CRP is not abnormal, it is designated as an HS-MGPS of 0.

## Outcome event

The primary endpoint event was MACEs during hospitalization, which included cardiovascular death, deterioration of heart failure, cardiogenic shock, mechanical complications of myocardial infarction, cerebral infarction, myocardial reinfarction and persistent ventricular arrhythmia. The secondary endpoint event was all-cause mortality during hospitalization.

## Statistical analysis

The data were analyzed using SPSS 25.0 (SPSS, Inc., Chicago, IL, USA). Univariate analysis of continuous variables with normal distribution and equal variance was performed by one-way ANOVA. One-to-one comparisons were performed with the Student-Newman–Keuls test. When the variance was uneven, Dunnett's test was used for one-to-one comparisons. Continuous variables of skewed distribution were first logarithmically transformed. If logarithmic data conformed to a normal distribution, the

analysis was the same, but if they still did not conform to orthodox distribution, the median and quartiles were taken and the Kruskal–Wallis test was used. The Pearson chi-square test or Fisher's exact test was used to analyze the rate according to the amount of data. Univariate/multivariate associations between clinical variables and in-hospital endpoints were estimated by logistic regression analysis with a forward stepwise logistic regression (LR) model. The clinical variables in univariate analysis ($P < 0.10$) and variables of clinical interest were included in multivariate analysis, but those variables that participated in the scoring system were excluded. In addition, in the logistic regression analysis, the variables with missing values were filled by the method of expectation maximization. The calibration of multivariate logistic regression model was evaluated by Hosmer–Lemeshow good of fit test. Receiver operating characteristic (ROC) curve analysis was performed, and the area under the curve (AUC) was calculated to determine the predictive ability of each score for the endpoint events. In addition, in subgroup analysis, the quantitative analysis of ROC curve comparison of three kinds of GPSs for MACEs prediction was completed by Delong method. The odds ratio (OR) (95% confidence interval (CI)) and AUC (95% CI), rate, rank sum number (RSN), median (quartile), mean ± standard error (SE) and mean ± standard deviation (SD) were taken as statistical values in corresponding cases. A bilateral $P < 0.05$ was considered statistically significant.

## Patient and public involvement

Participants were not involved in the study design, recruitment, implementation, article writing or data collection. Patients did not incur additional medical burden in the study. The results of the study will be disseminated to all patients and medical institutions through academic conferences, news reports and health publicity.

## RESULTS

### Baseline patient characteristics

A total of 188 patients were included in the study (Fig. 1). The main clinical features of the patients are shown in Table 1. In summary, the average age was 68.21 ± 14.89 years, 77.7% of patients were male and 71.8%, 42.6% and 38.8% of patients had hypertension, diabetes mellitus and HTI-GPS, respectively. Of these patients, 60 (31.9%) had a MACE during hospitalization and 19 (10.1%) had all-cause mortality during hospitalization. The baseline characteristics of patients were classified according to the grades of the three GPSs. Compared with the low-score group, the high-score group had older patients ($P < 0.001$), a higher prevalence of diabetes ($P < 0.05$), a higher HTI-GPS rate ($P < 0.001$), a lower diastolic blood pressure ($P < 0.05$), higher Killip classes ($P < 0.001$), lower hemoglobin levels ($P < 0.001$), higher D-dimer levels ($P < 0.001$), higher creatinine levels ($P < 0.001$) and higher BNP levels ($P < 0.001$).

### Prediction of the primary endpoint event

The results of the ROC curve analysis for the incidence of MACE during hospitalization showed that the AUC value of the GPS (0.820 (95% CI [0.754–0.885]), $P < 0.001$) was higher than that of the MGPS (0.789 (95% CI [0.715–0.863]), $P < 0.001$)

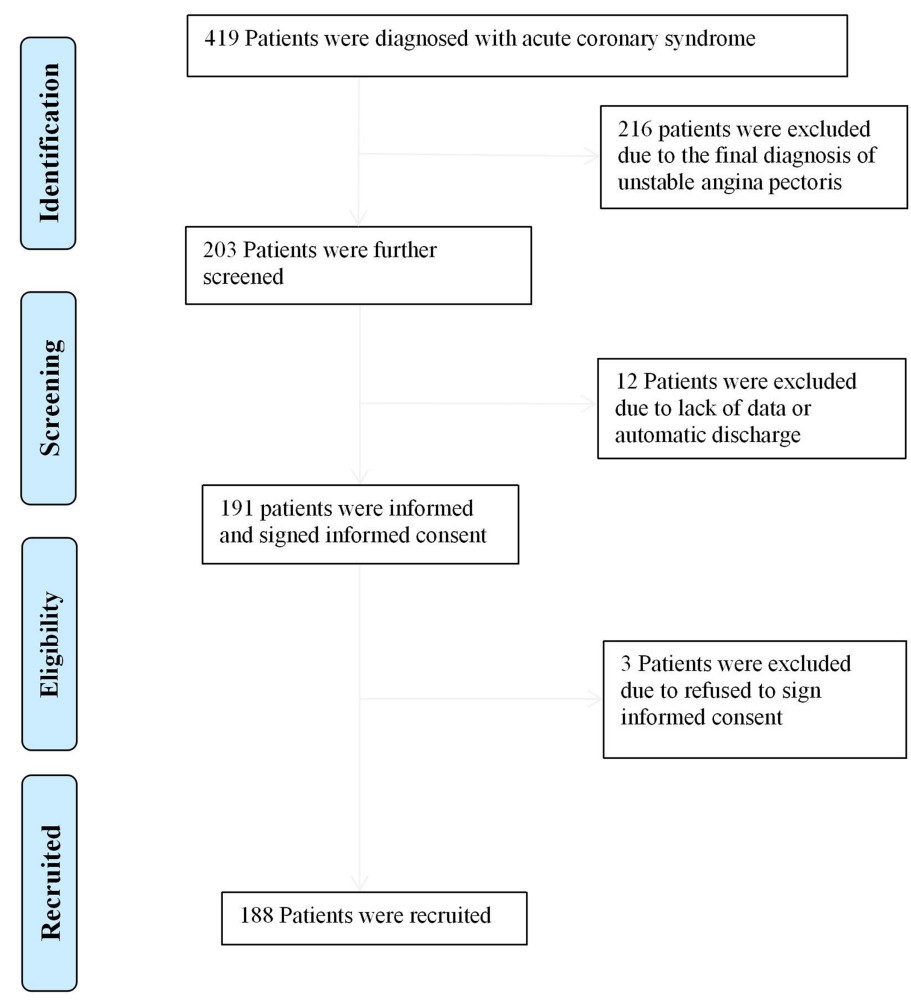

**Figure 1 Flow diagram for recruitment of patients.**

(Fig. 2A), HS-MGPS (0.787 (95% CI [0.717–0.856]), *P* < 0.001), GRACE score (0.743 (95% CI [0.672–0.814]), *P* < 0.001), CK-MB level (0.485 (95% CI [0.394–0.576]), *P* = 0.745), TNI level (0.593 (95% CI [0.448–0.629]), *P* = 0.394), LDL level (0.440 (95% CI [0.346–0.534]), *P* = 0.193), and BNP level (0.764 (95% CI [0.688–0.840]), *P* < 0.001). The predictive value of the H-CRP level (0.809 (95% CI [0.747–0.871]), *P* < 0.001) and albumin level (0.796 (95% CI [0.724–0.869]), *P* < 0.001) was also lower than that of the GPS.

## Prediction of the secondary endpoint event

The results of the ROC curve analysis for the incidence of all-cause mortality during hospitalization indicated that the AUC value of the GPS (0.696 (95% CI [0.561–0.831]), *P* = 0.005) was similar to that of the HS-MGPS (0.698 (95% CI [0.569–0.826]), *P* = 0.005) (Fig. 2B) and BNP level (0.690 (95% CI [0.562–0.818]), *P* = 0.009) and larger than that of the MGPS (0.668 (95% CI [0.525–0.812]), *P* = 0.016), CK-MB level (0.461 (95% CI [0.320–0.601]), *P* = 0.574), TNI level (0.527 (95% CI [0.389–0.665]), *P* = 0.701), LDL

**Table 1 Relationships between clinical characteristics and the GPSs in patients with acute myocardial infarction.**

| Vairable | GPS | | | | MGPS | | | | HS-MGPS | | | |
|---|---|---|---|---|---|---|---|---|---|---|---|---|
| | 0 | 1 | 2 | P | 0 | 1 | 2 | P | 0 | 1 | 2 | P |
| Age, (years) | 62.49 ± 15.05 | 70.02 ± 13.68 | 77.06 ± 10.43 | <0.001 | 63.83 ± 15.10 | 69.03 ± 14.11 | 77.06 ± 10.43 | <0.001 | 63.51 ± 16.32 | 64.95 ± 14.58 | 68.21 ± 14.89 | <0.001 |
| Males, (n, %) | 80 (54.8) | 31 (21.2) | 35 (24.0) | 0.010 | 90 (61.6) | 21 (14.4) | 35 (24.0) | 0.049 | 37 (25.3) | 68 (46.6) | 41 (28.1) | 0.022 |
| Hypertension, (n, %) | 62 (45.9) | 33 (24.4) | 40 (29.6) | 0.280 | 75 (55.6) | 20 (14.8) | 40 (29.6) | 0.269 | 25 (18.5) | 60 (44.4) | 50 (37.0) | 0.059 |
| Diabetics, (n, %) | 31 (38.8) | 20 (25.0) | 29 (36.3) | 0.020 | 38 (47.5) | 13 (16.3) | 29 (36.3) | 0.029 | 16 (20.0) | 30 (37.5) | 34 (42.5) | 0.037 |
| HTI-GPS, (n, %) | 8 (11.0) | 23 (31.5) | 42 (57.5) | <0.001 | 13 (17.8) | 18 (24.7) | 42 (57.5) | <0.001 | 3 (4.1) | 23 (31.5) | 47 (64.4) | <0.001 |
| Heart rate, (times/min) | 78.27 ± 15.80 | 83.58 ± 18.53 | 82.66 ± 20.11 | 0.174 | 78.96 ± 16.58 | 83.77 ± 17.61 | 82.66 ± 20.11 | 0.174 | 74.02 ± 11.34 | 82.28 ± 17.47 | 83.05 ± 20.66 | 0.022 |
| SBP, (mm Hg) | 129.65 ± 23.65 | 132.02 ± 26.15 | 130.46 ± 29.60 | 0.882 | 130.75 ± 24.68 | 129.30 ± 23.90 | 130.46 ± 29.60 | 0.882 | 129.73 ± 25.10 | 130.38 ± 23.07 | 131.00 ± 30.03 | 0.971 |
| DBP, (mm Hg) | 76.16 ± 12.09 | 75.33 ± 14.08 | 69.28 ± 12.92 | 0.008 | 75.89 ± 12.25 | 75.90 ± 14.56 | 69.28 ± 12.92 | 0.009 | 74.59 ± 11.65 | 76.94 ± 13.19 | 69.89 ± 12.92 | 0.005 |
| Killip classification, (RSN) | 75.34 | 105.04 | 120.06 | <0.001 | 81.88 | 96.82 | 120.06 | <0.001 | 79.55 | 83.62 | 119.89 | <0.001 |
| EF (Simpson), (×100%) | 45.09 ± 8.49 | 45.80 ± 13.52 | 46.67 ± 8.71 | 0.956 | 45.09 ± 8.49 | 45.80 ± 13.52 | 46.67 ± 8.71 | 0.956 | 45.00 ± 5.66 | 45.40 ± 1.08 | 46.67 ± 8.71 | 0.962 |
| EF(M), (×100%) | 60.47 ± 7.93 | 57.40 ± 11.69 | 56.61 ± 10.09 | 0.211 | 60.97 ± 8.02 | 54.73 ± 11.63 | 56.61 ± 10.09 | 0.035 | 60.97 ± 5.65 | 58.78 ± 10.16 | 57.31 ± 10.19 | 0.485 |
| Hemoglobin, (g/L) | 134.82 ± 23.84 | 124.30 ± 25.72 | 111.16 ± 26.64 | <0.001 | 133.00 ± 24.79 | 125.48 ± 24.74 | 111.16 ± 26.64 | <0.001 | 133.71 ± 24.49 | 132.52 ± 25.72 | 111.48 ± 26.12 | <0.001 |
| Platelet, (×10^12/L) | 207.58 ± 55.36 | 202.41 ± 76.83 | 192.44 ± 74.47 | 0.433 | 203.54 ± 57.78 | 213.84 ± 79.36 | 192.44 ± 74.47 | 0.356 | 198.10 ± 58.83 | 213.31 ± 62.77 | 189.56 ± 74.07 | 0.091 |
| D-Dimer, (mg/L) | 0.68 ± 1.01 | 1.46 ± 1.81 | 4.86 ± 8.44 | <0.001 | 0.78 ± 1.02 | 1.51 ± 2.14 | 4.86 ± 8.44 | <0.001 | 0.74 ± 1.15 | 0.98 ± 1.50 | 4.23 ± 7.76 | <0.001 |
| ALT, (U/L) | 45.64 ± 28.57 | 84.39 ± 225.13 | 153.54 ± 338.14 | 0.014 | 44.21 ± 28.00 | 108.06 ± 272.03 | 153.54 ± 338.14 | 0.008 | 43.27 ± 28.29 | 69.35 ± 165.83 | 131.48 ± 309.34 | 0.080 |
| CK, (U/L) | 1,292 ± 1,409 | 1,259 ± 1,651 | 1,065 ± 2,179 | 0.741 | 1,231 ± 1,347 | 1,455 ± 1,915 | 1,065 ± 2,179 | 0.605 | 1,335 ± 1,659 | 1,292 ± 1,476 | 1,052 ± 1,999 | 0.625 |
| CK-MB, (U/L) | 146.22 ± 155.98 | 118.50 ± 119.43 | 129.20 ± 257.57 | 0.677 | 138.37 ± 148.23 | 132.17 ± 135.33 | 129.20 ± 257.57 | 0.954 | 145.99 ± 167.96 | 137.75 ± 140.19 | 123.46 ± 235.22 | 0.813 |
| LDL, (mmol/L) | 3.07 ± 1.03 | 3.03 ± 0.89 | 2.70 ± 1.12 | 0.115 | 3.04 ± 0.98 | 3.12 ± 0.99 | 2.70 ± 1.12 | 0.109 | 2.96 ± 0.95 | 3.12 ± 1.04 | 2.74 ± 1.03 | 0.090 |
| Creatinine, (umol/L) | 107.99 ± 130.32 | 160.39 ± 179.50 | 240.24 ± 222.54 | <0.001 | 117.46 ± 144.97 | 153.06 ± 165.47 | 240.24 ± 222.54 | <0.001 | 96.98 ± 89.67 | 127.83 ± 156.28 | 235.34 ± 223.48 | <0.001 |
| BNP, (pg/mL) | 321.89 ± 427.15 | 859.82 ± 1,226.71 | 1,767.32 ± 1,555.29 | <0.001 | 345.03 ± 457.71 | 1,056.86 ± 1,411.58 | 1,767.32 ± 1,555.29 | <0.001 | 235.44 ± 281.82 | 605.84 ± 978.26 | 1,562.49 ± 1,502.86 | <0.001 |
| TNI, (ng/mL) | 29.65 ± 33.42 | 27.66 ± 34.39 | 21.32 ± 28.58 | 0.341 | 29.14 ± 33.57 | 28.46 ± 33.39 | 21.32 ± 28.58 | 0.359 | 31.22 ± 34.43 | 27.43 ± 32.82 | 23.41 ± 30.72 | 0.486 |

**Note:**

Relationships between clinical characteristics and the GPSs in patients with acute myocardial infarction. GPSs, Inflammation-based Glasgow Prognostic Scores; GPS, inflammation-based Glasgow Prognostic Score; MGPS, modified inflammation-based Glasgow Prognostic Score; HS-MGPS, high-sensitivity CRP-modified inflammation-based Glasgow Prognostic Score; HTI-GPS, high tendency interference GPS group; SBP, systolic blood pressure; DBP, diastolic blood pressure; RSN, rank sum number; EF, ejection fraction; ALT, Alanine aminotransferase; CK, creatine kinase; CK-MB, creatine kinase MB; LDL, low density lipoproteincreatinine; BNP, type B natriuretic peptide; TNI, troponin I; SD, standard deviation. Except for rate and RSN, the rest are represented as mean ± SD.

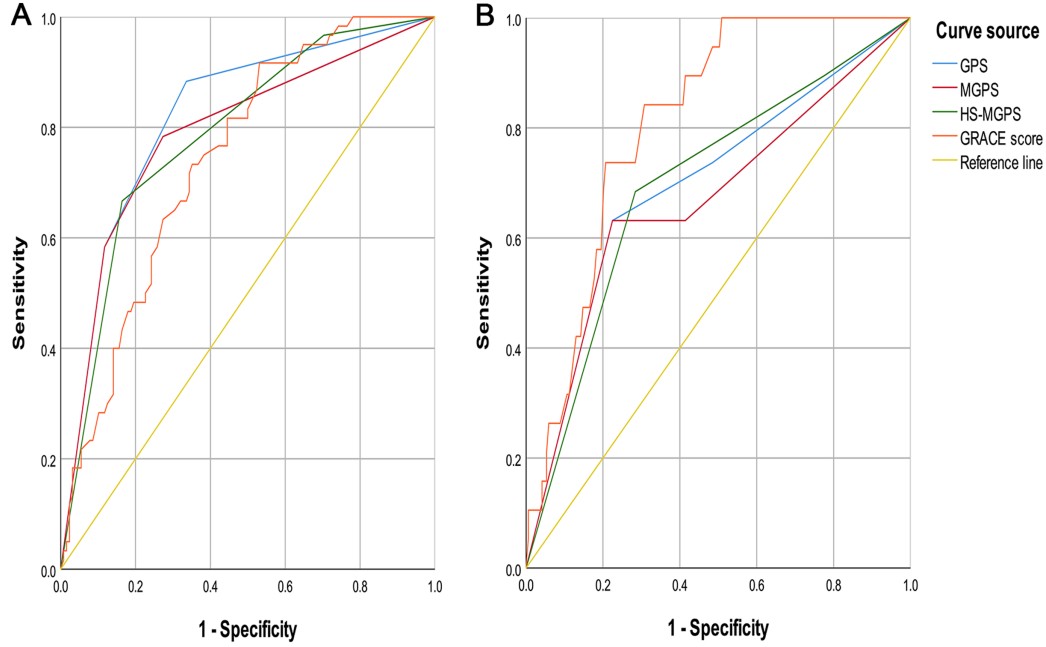

**Figure 2  ROC Curves of GPSs and GRACE score.** (A) Endpoint event with MACE. (B) Endpoint event with all-cause mortality. ROC, receiver operating characteristic; GPSs, Inflammation-based Glasgow Prognostic Scores; GRACE, Global Registry of Acute Coronary Events; MACE, Global Registry of Acute Coronary Events; GPS, inflammation-based Glasgow Prognostic Score; MGPS, modified inflammation-based Glasgow Prognostic Score; HS-MGPS, high-sensitivity CRP-modified inflammation-based Glasgow Prognostic Score; AUC, area under the curve; CI, confidence interval. All values are represented as AUC (95% CI).               

level (0.430 (95% CI [0.269–0.592]), $P$ = 0.332), and H-CRP level (0.682 (95% CI [0.558–0.806]), $P$ = 0.009) but smaller than that of the GRACE score (0.812 (95% CI [0.734–0.889]), $P$ < 0.001) and albumin level (0.724 (95% CI [0.585–0.863]), $P$ = 0.001).

## Analysis of the different levels of GPSs

Patients with a GPS of 2 had higher GRACE scores than those with a score of 1 (17.15 ± 6.01, $P$ = 0.017) and 0 (30.85 ± 4.47, $P$ < 0.001) (Fig. 3A). Patients with an MGPS of 2 also had higher GRACE scores than those with a score of 1 (19.27 ± 7.60, $P$ = 0.044) and 0 (28.40 ± 4.28, $P$ < 0.001) (Fig. 3B). Similarly, patients with an HS-MGPS of 2 also had higher GRACE scores than those with a score of 1 (24.49 ± 4.63, $P$ < 0.001) and 0 (28.12 ± 5.94, $P$ < 0.001) (Fig. 3C).

## Logistic regression analysis to evaluate the GPS risk in endpoint events

The missing rates of LDL, D-dimer, BNP, EF (Simpson) and EF (M) were 1.1%, 1.1%, 13.8%, 89.4% and 52.1% respectively(see File S2). And in multivariate analysis, M-type method was used as EF measurement results to reduce the impact of too many missing values on the outcome.

Univariate logistic regression analysis showed that the incidence of MACE during hospitalization was positively correlated with the GPS ($P$ < 0.001), and the GRACE score

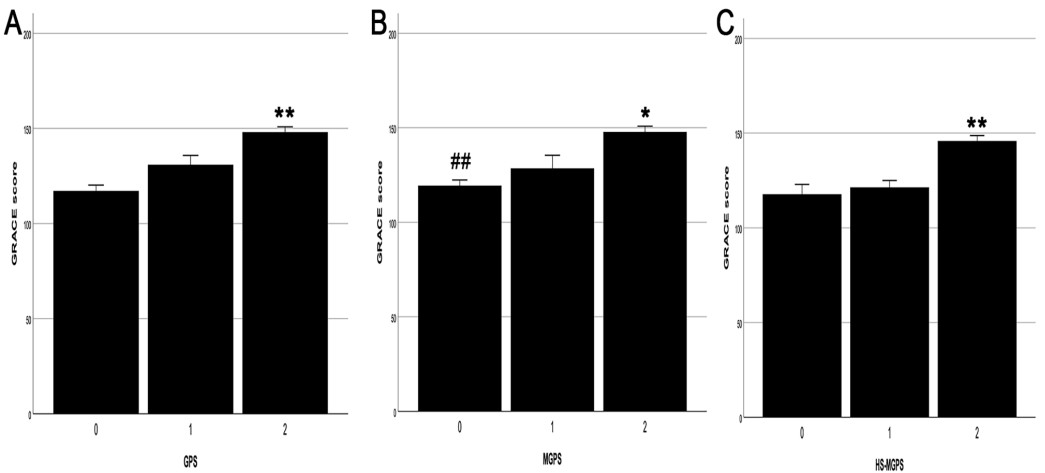

**Figure 3 Relationship between the grade of GPSs and GRACE score.** (A) The grade of GPS and GRACE score. (B) The grade of MGPS and GRACE score. (C) The grade of HS-MGPS and GRACE score. GPSs, Inflammation-based Glasgow Prognostic Scores; GRACE, Global Registry of Acute Coronary Events; GPS, inflammation-based Glasgow Prognostic Score; MGPS, modified inflammation-based Glasgow Prognostic Score; HS-MGPS, high-sensitivity CRP-modified inflammation-based Glasgow Prognostic Score; SE, standard error. All values are represented as mean ± SE. $^{**}P < 0.001$ vs. 0 or 1 score group. $^{*}P < 0.05$ vs. 1 score gourp. $^{##}P < 0.001$ vs. 2 score group.

was also significant ($P < 0.001$) (Table 2). Multivariate logistic regression analysis showed that the GPS was an independent risk factor for the incidence of MACE during hospitalization and the model had an adequate calibration ($P_{Hosmer–Lemeshow} = 0.830$). Compared with the OR value for a GPS of 0, the OR for a GPS of 1 was 7.173 (95% CI [2.425–21.216]), $P < 0.001$) and that for a GPS of 2 was 18.636 (95% CI [5.813–59.746]), $P < 0.001$). The GRACE score was also an independent risk factor for the incidence of MACE during hospitalization (1.019 (95% CI [1.004–1.035]), $P = 0.015$). In addition, BNP and TNI levels were also considered independent risk factors for MACE.

Univariate logistic regression analysis showed that all-cause mortality during hospitalization was positively correlated with the GPS ($P = 0.002$), and the GRACE score was also significant ($P < 0.001$) (Table 3). However, multivariate logistic regression analysis ($P_{Hosmer–Lemeshow} = 0.351$) showed that the GPS was not an independent risk factor for all-cause mortality during hospitalization ($P = 0.302$), but the GRACE score was (1.040 (95% CI [1.017–1.064]), $P = 0.001$). In addition, the type of PCI was shown to be an independent risk factor for all-cause mortality during hospitalization (0.236 (95% CI [0.089–0.625]), $P = 0.004$).

## Subgroup analysis

Subgroups were divided according to the type of AMI, the presence of disease interference GPSs and the type of PCI. The prediction of MACE during hospitalization was assessed by ROC curve analysis. In the STEMI group, the AUC value of the GPS (0.787 (95% CI [0.688–0.887]), $P < 0.001$) seemed to be larger than that of the MGPS (0.740 (95% CI [0.628–0.852]), $P < 0.001$) and HS-MGPS (0.758 (95% CI [0.656–0.861]), $P < 0.001$)

**Table 2 Logistic regression analysis of MACE during hospitalization.**

| Vairable | OR | Univariate analysis for 95% CI | P | OR | Multivariate analysis for 95% CI | P |
|---|---|---|---|---|---|---|
| GPS | | | <0.001 | | | <0.001 |
| GPS (1 vs 0) | 7.806 | [2.954–20.631] | <0.001 | 7.173 | [2.425–21.216] | <0.001 |
| GPS (2 vs 0) | 28.333 | [10.637–75.471] | <0.001 | 18.636 | [5.813–59.746] | <0.001 |
| MGPS | | | <0.001 | | | – |
| MGPS (1 vs 0) | 4.292 | [1.708–10.786] | 0.002 | | | – |
| MGPS (2 vs 0) | 16.692 | [7.219–38.598] | <0.001 | | | – |
| HS-MGPS | | | <0.001 | | | – |
| HS-MGPS (1 vs 0) | 4.975 | [1.091–22.517] | 0.038 | | | – |
| HS-MGPS (2 vs 0) | 36.190 | [7.940–164.948] | <0.001 | | | – |
| GRACE score | 1.033 | [1.020–1.046] | <0.001 | 1.019 | [1.004–1.035] | 0.015 |
| Age | 1.070 | [1.040–1.100] | <0.001 | | | – |
| Males | 0.538 | [0.265–1.094] | 0.087 | | | 0.746 |
| Hypertension | 0.990 | [0.501–1.956] | 0.976 | | | 0.291 |
| Diabetics | 2.582 | [1.377–4.841] | 0.003 | | | 0.079 |
| HTI-GPS | 4.590 | [2.388–8.820] | <0.001 | | | 0.428 |
| PCI type | 0.241 | [0.126–0.463] | <0.001 | | | 0.062 |
| Heart rate | 1.035 | [1.016–1.055] | <0.001 | | | – |
| SBP | 1.001 | [0.990–1.013] | 0.811 | | | – |
| DBP | 0.969 | [0.944–0.993] | 0.013 | | | 0.910 |
| Killip class | 6.534 | [3.599–11.861] | <0.001 | | | – |
| EF (Simpson) | 0.001 | [0.000–0.070] | 0.001 | | | – |
| EF (M) | 0.002 | [0.000–0.183] | 0.007 | | | 0.655 |
| Hemoglobin | 0.973 | [0.960–0.986] | <0.001 | | | 0.114 |
| Platelet | 0.993 | [0.988–0.998] | 0.009 | | | 0.133 |
| D-dimer | 1.294 | [1.099–1.523] | 0.002 | | | 0.501 |
| ALT | 1.002 | [1.000–1.014] | 0.036 | | | 0.759 |
| CK | 1.000 | [1.000–1.000] | 0.626 | | | 0.452 |
| CK-MB | 1.001 | [0.999–1.002] | 0.394 | | | 0.805 |
| Albumin | 0.747 | [0.676–0.824] | <0.001 | | | – |
| HS-CRP | 1.015 | [1.009–1.022] | <0.001 | | | – |
| LDL | 0.883 | [0.649–1.203] | 0.431 | | | 0.706 |
| Creatinine | 1.003 | [1.001–1.005] | <0.001 | | | 0.336 |
| BNP | 1.001 | [1.000–1.001] | <0.001 | 1.000 | [1.000–1.001] | 0.035 |
| TNI | 1.006 | [0.997–1.016] | 0.184 | 1.020 | [1.007–1.033] | 0.003 |

**Note:**
Logistic regression analysis of MACE during hospitalization. OR, odds ratio; CI, confidence interval; MACE, Global Registry of Acute Coronary Events; GPS, inflammation-based Glasgow Prognostic Score; MGPS, modified inflammation-based Glasgow Prognostic Score; HS-MGPS, high-sensitivity CRP-modified inflammation-based Glasgow Prognostic Score; GRACE, Global Registry of Acute Coronary Events; PCI, percutaneous coronary intervention; HTI-GPS, high tendency interference GPS group; SBP, systolic blood pressure; DBP, diastolic blood pressure; EF, ejection fraction; ALT, Alanine aminotransferase; CK, creatine kinase; CK-MB, creatine kinase MB; LDL, low density lipoproteincreatinine; BNP, type B natriuretic peptide; TNI, troponin I.
**Table 3 Logistic regression analysis of all-cause mortality during hospitalization.**

| Vairable | OR | Univariate analysis for 95% CI | P | OR | Multivariate analysis for 95% CI | P |
|---|---|---|---|---|---|---|
| GPS | | | 0.002 | | | 0.302 |
| GPS (1 vs 0) | 0.791 | [0.147–4.241] | 0.784 | | | 0.134 |
| GPS (2 vs 0) | 5.495 | [1.810–16.685] | 0.003 | | | 0.253 |
| MGPS | | | 0.014 | | | – |
| MGPS (1 vs 0) | 0.000 | | 0.998 | | | – |
| MGPS (2 vs 0) | 4.466 | [1.636–12.194] | 0.013 | | | – |
| HS-MGPS | | | 0.005 | | | – |
| HS-MGPS (1 vs 0) | 0.916 | [0.161–5.218] | 0.921 | | | – |
| HS-MGPS (2 vs 0) | 5.146 | [1.094–24.204] | 0.038 | | | – |
| GRACE score | 1.044 | [1.022–1.066] | <0.001 | 1.040 | [1.017–1.064] | 0.001 |
| Age | 1.107 | [1.048–1.170] | <0.001 | | | – |
| Males | 0.785 | [0.265–2.321] | 0.661 | | | 0.808 |
| Hypertension | 1.111 | [0.379–3.253] | 0.848 | | | 0.908 |
| Diabetics | 1.242 | [0.480–3.215] | 0.655 | | | 0.813 |
| HTI-GPS | 3.936 | [1.423–10.887] | 0.008 | | | 0.499 |
| PCI type | 0.155 | [0.060–0.404] | <0.001 | 0.236 | [0.089–0.625] | 0.004 |
| Heart rate | 1.027 | [1.002–1.052] | 0.032 | | | – |
| SBP | 0.991 | [0.972–1.010] | 0.331 | | | – |
| DBP | 0.956 | [0.919–0.996] | 0.029 | | | 0.696 |
| Killip class | 1.928 | [1.214–3.060] | 0.005 | | | – |
| EF (Simpson) | 0.000 | [0.000–0.033] | 0.002 | | | – |
| EF (M) | 0.003 | [0.000–0.814] | 0.042 | | | 0.328 |
| Hemoglobin | 0.980 | [0.964–0.997] | 0.019 | | | 0.907 |
| Platelet | 0.997 | [0.990–1.005] | 0.462 | | | 0.309 |
| D-Dimer | 1.113 | [1.027–1.205] | 0.009 | | | 0.098 |
| ALT | 1.001 | [0.999–1.002] | 0.584 | | | 0.644 |
| CK | 1.000 | [1.000–1.000] | 0.583 | | | 0.707 |
| CK-MB | 1.000 | [0.998–1.003] | 0.945 | | | 0.358 |
| Albumin | 0.802 | [0.716–0.898] | <0.001 | | | – |
| HS-CRP | 1.010 | [1.003–1.017] | 0.004 | | | – |
| LDL | 0.908 | [0.560–1.471] | 0.649 | | | 0.149 |
| Creatinine | 1.001 | [0.999–1.003] | 0.224 | | | 0.606 |
| BNP | 1.000 | [1.000–1.001] | 0.004 | | | 0.541 |
| TNI | 1.004 | [0.990–1.018] | 0.590 | | | 0.412 |

**Note:**
Logistic regression analysis of all-cause mortality during hospitalization. OR, odds ratio; CI, confidence interval; MACE, Global Registry of Acute Coronary Events; GPS, inflammation-based Glasgow Prognostic Score; MGPS, modified inflammation-based Glasgow Prognostic Score; HS-MGPS, high-sensitivity CRP-modified inflammation-based Glasgow Prognostic Score; GRACE, Global Registry of Acute Coronary Events; PCI, percutaneous coronary intervention; HTI-GPS, high tendency interference GPS group; SBP, systolic blood pressure; DBP, diastolic blood pressure; EF, ejection fraction; ALT, Alanine aminotransferase; CK, creatine kinase; CK-MB, creatine kinase MB; LDL, low density lipoproteincreatinine; BNP, type B natriuretic peptide; TNI, troponin I. All values are represented as OR (95% CI).

**Table 4  ROC analysis of in-hospital MACE for subgroups.**

| Subgroup | GPS | | | MGPS | | | HS-MGPS | | |
|---|---|---|---|---|---|---|---|---|---|
| | AUC | CI | *P* | AUC | CI | *P* | AUC | CI | *P* |
| STEMI | 0.787 | [0.688–0.887] | <0.001 | 0.740 | [0.628–0.852] | <0.001 | 0.758 | [0.656–0.861] | <0.001 |
| NSTEMI | 0.855 | [0.772–0.938] | <0.001 | 0.838 | [0.746–0.931] | <0.001 | 0.812 | [0.719–0.906] | <0.001 |
| HTI-GPS | 0.711 | [0.591–0.832] | 0.002 | 0.717 | [0.596–0.837] | 0.001 | 0.661 | [0.534–0.787] | 0.018 |
| LTI-GPS | 0.797 | [0.679–0.915] | <0.001 | 0.702 | [0.563–0.841] | 0.003 | 0.765 | [0.647–0.883] | <0.001 |
| PPCI | 0.800 | [0.707–0.893] | <0.001 | 0.756 | [0.649–0.863] | <0.001 | 0.775 | [0.682–0.868] | <0.001 |
| EPCI | 0.750 | [0.442–1.000] | 0.134 | 0.783 | [0.493–1.000] | 0.089 | 0.700 | [0.419–0.981] | 0.230 |
| Non-PCI | 0.716 | [0.532–0.900] | 0.030 | 0.692 | [0.509–0.876] | 0.053 | 0.714 | [0.533–0.896] | 0.031 |

**Note:**
ROC analysis of in-hospital MACE for subgroups. ROC, receiver operating characteristic; MACE, Global Registry of Acute Coronary Events; GPS, inflammation-based Glasgow Prognostic Score; MGPS, modified inflammation-based Glasgow Prognostic Score; HS-MGPS, high-sensitivity CRP-modified inflammation-based Glasgow Prognostic Score. AUC, area under the curve; CI, confidence interval; STEMI, ST-segment elevation myocardial infarction; NSTEMI, non-ST-segment elevation myocardial infarction; HTI-GPS, high tendency interference GPS group; LTI-GPS, low tendency interference GPS group; PCI, percutaneous coronary intervention; PPCI, primary PCI; EPCI, elective PCI. All values are represented as AUC (95% CI).

(Table 4), but Delong test showed no significant statistical difference in the predictive value of the three scores (see File S2). In the NSTEMI group, the AUC value of the GPS (0.855 (95% CI [0.772–0.938]), *P* < 0.001) also seemed to be larger than that of the MGPS (0.838 (95% CI [0.746–0.931]), *P* < 0.001) and HS-MGPS (0.812 (95% CI [0.719–0.906]), *P* < 0.001), but Delong test showed no significant statistical difference in the predictive value of the three scores. In the HTI-GPS group, the AUC value of the GPS (0.711 (95% CI [0.591–0.832]), *P* = 0.002) was similar to that of the MGPS (0.717 (95% CI [0.596–0.837]), *P* = 0.001) but seemed to be larger than that of the HS-MGPS (0.661 (95% CI [0.534–0.787]), *P* = 0.018). However, Delong test s till showed that there was no significant statistical difference. In the LTI-GPS group, the AUC value of the GPS (0.797 (95% CI [0.679–0.915]), *P* < 0.001) seemed to be larger than that of the MGPS (0.702 (95% CI [0.563–0.841]), *P* = 0.003) and HS-MGPS (0.765 (95% CI [0.647–0.883]), *P* < 0.001), and Delong test showed that the predictive value of GPS was statistically different from that of MGPS. In the PPCI group, the AUC value of the GPS (0.800 (95% CI [0.707–0.893]), *P* < 0.001) seemed to be larger than that of the MGPS (0.756 (95% CI [0.649–0.863]), *P* < 0.001) and HS-MGPS (0.775 (95% CI [0.682–0.868]), *P* < 0.001), but Delong test showed that there was no statistical difference. In the EPCI group, the ROC curve analysis of the three GPSs did not reach statistical significance. In the non-PCI group (conservative drug therapy), the AUC value of the GPS (0.716 (95% CI [0.532–0.900]), *P* = 0.030) was similar to that of the HS-MGPS (0.714 (95% CI [0.533–0.896]), *P* = 0.031) but seemed to be larger than that of the MGPS (0.692 (95% CI [0.509–0.876]), *P* = 0.053), but Delong test still showed that there was no significant statistical difference.

## DISCUSSION

This study investigated the predictive value of the three GPSs for the incidence of MACE and all-cause mortality in patients with AMI during hospitalization. The main findings are as follows: (1) The predictive ability of the GPS for the incidence of MACE during

hospitalization was greater than that of the MGPS, HS-MGPS and GRACE score, and multiple logistic regression analysis showed that the GPS was an independent risk factor for the occurrence of MACE, (2) The three GPSs were less able to predict all-cause mortality during hospitalization than the GRACE score and although they were risk factors for all-cause mortality in univariate logistic regression analysis, they were not independent risk factors for all-cause mortality in multiple logistic regression analysis, (3) In the subgroups classified according to the type of AMI, the presence of disease interference GPSs and the type of PCI, the ability of the GPS to predict the occurrence of MACE was generally greater than that of the MGPS and HS-MGPS.

In the long course of research on the pathogenesis of coronary atherosclerotic heart disease, the theory of inflammation, as one of the three theories, was first proposed by *Virchow (1856)*. They believe that atherosclerosis is an inflammation of the intima of the arteries but is different from common inflammation. It starts with the deposition of CRP in the local area of atherosclerosis, which induces endothelial cells to secrete and express adhesion molecules and chemokines, promotes macrophages to express cytokines and tissue factors and stimulates the uptake of LDL. It also stimulates macrophages to produce prothrombotic factors and endothelial cells and stimulates monocytes and lymphocytes to produce oxygen free radicals, which induces nuclear transcription factor-mediated arterial inflammation (*Linden et al., 2014*; *Lorenzatti & Retzlaff, 2016*; *Ridker & Luscher, 2014*; *Kottoor & Arora, 2018*). The Centers for Disease Control and Prevention and the American Heart Association (AHA) also suggested that CRP is one of the strongest predictors of cardiovascular disease. CRP of 3–10 mg/L suggests a high risk of AMI (*Pearson et al., 2003*). In addition to maintaining plasma osmotic pressure and acting as a carrier, albumin is also considered to be an important extracellular antioxidant (*Kawai et al., 2018*; *Rosas-Diaz et al., 2015*). The normal concentration of albumin may play an important role in inhibiting platelet activation and the apoptosis of vascular endothelial cells (*Lam et al., 2013*; *Fernandez et al., 2019*; *Anraku et al., 2013*). Therefore, GPSs are structurally based on inflammation theory to predict the occurrence of MACE in patients with AMI.

In this study, we included three kinds of GPSs. The GPS first proposed by *Forrest et al. (2003)* was found to be able to predict the survival time of patients with non-small cell lung cancer. Recent studies suggested that it has predictive value for the survival time of STEMI patients undergoing PPCI (*Jia et al., 2018*; *Wang et al., 2019*). The MGPS based on the GPS highlights the importance of CRP. The MGPS has been more widely used to assess the prognosis of patients with cancer than the GPS (*Wu et al., 2018*; *Sato et al., 2019*), but the prognostic evaluation ability of patients with AMI is still unclear. In addition, the HS-MGPS based on the MGPS further highlights the importance of CRP, since the cutoff setting of CRP is equivalent to that in the high-risk group of AMI according to the guidelines of the AHA (*Pearson et al., 2003*), which also implies that this score may be more accurate than the first two GPSs. However, our study found that the accuracy of the GPS in predicting MACE and all-cause mortality in patients with AMI during hospitalization was higher than that of the MGPS and HS-MGPS. In the subgroup analysis, in addition to the HTI-GPS, EPCI and conservative drug therapy groups, the ability of the

GPS to predict MACE was still higher than that of the MGPS and HS-MGPS. The main difference between the GPS and the two improved scores is whether albumin can independently occupy 1 score in the GPS system, that is, the importance of albumin. In addition to the aforementioned inflammatory hypothesis based on albumin, patients with AMI are in a state of stress, with a significant increase in the basal metabolic rate, an increase in energy demand, changes in energy consumption pathways during stress, a limited use of glucose, and the use of albumin as an important energy source. The body relies on a large amount of protein decomposition to obtain energy, so the amount of serum albumin consumption also indirectly reflects the extent of AMI regarding the overall loss in patients. Studies have suggested that albumin levels are negatively correlated with the incidence of AMI (*He et al., 2016*; *Djousse et al., 2002*), and in patients with acute coronary syndrome (ACS), low albumin levels were found to be independent predictors of all-cause mortality and the deterioration of heart failure during hospitalization (*Xia et al., 2018*; *Kurtul et al., 2016*; *Gonzalez-Pacheco et al., 2017*). Therefore, the levels of albumin and CRP may be independent and equally important in predicting the prognosis of patients with AMI.

The secondary endpoint results showed that the three GPSs were less effective than the GRACE score in predicting all-cause mortality in patients with AMI during hospitalization. This may be due to the inclusion of patients in the real world who have many other diseases, such as diabetes, hypertension and kidney disease. As a result, conditions affecting all-cause mortality can become very unpredictable and not limited to cardiovascular diseases based on the inflammation hypothesis. However, the components of the GRACE score include heart rate and systolic blood pressure as references. These values, as basic vital signs, can reflect the patient's response to all acute and severe diseases. Therefore, the predictive value of the GRACE score in predicting all-cause mortality is significantly higher than that of the GPSs. Considering the results of this study, our team speculated that although the GRACE score was better than the GPSs in predicting all-cause mortality, the GRACE score was also more affected by "interference factors" than the GPSs, so it may cause more false positives in predicting MACE, making its predictive ability less than that of the GPSs. Both the American College of Cardiology/AHA and the European Society of Cardiology guidelines recommend the GRACE score as one of the main criteria for the risk assessment of patients with ACS (*Anderson et al., 2007*; *Bassand et al., 2007*). It has a high predictive ability for hospitalization and provides a 5-year mortality risk for patients with ACS (*Littnerova et al., 2015*; *Fox et al., 2010*). Therefore, the GRACE score is considered to be an important predictor of mortality in ACS. However, there is no inflammatory component in its composition. According to the results of this study, if the GRACE score and GPS are combined, their prognostic value for patients with AMI may be improved.

The results of subgroup analysis showed that the predictive ability of the GPS for MACE occurrence seemed to be greater than that of the MGPS and HS-MGPS through ROC curve analysis in both the STEMI and NSTEMI groups, but due to the limitation of sample size, there was no statistical difference in quantitative comparison among these subgroups. There were many clinical factors that affect CRP and albumin levels, but the

GPS had a good ability to predict MACE regardless of the presence of disease interference GPSs. In addition, in the PPCI and conservative drug therapy groups, the GPS still seemed to have a best ability to predict the occurrence of MACE compared with the modified GPSs, while in the EPCI group, none of the GPSs had meaningful predictive ability, but this may be due to the lack of data.

Patients with higher GPS scores may be accompanied by higher MACEs incidence than patients with lower scores. Strengthening the secondary prevention strategy of coronary heart disease and regular follow-up may be a good choice to prevent MACEs. In addition, anti-inflammatory treatment in the treatment strategy of coronary heart disease continues to experience a positive and negative process. In recent three large clinical trials, COLCOT study (*Tardif et al., 2019*) and CANTOS study (*Ridker et al., 2017*) showed the effectiveness and acceptability of anti-inflammatory treatment, but CIRT study (*Ridker et al., 2019*) showed that anti-inflammatory treatment not only failed to reach positive end points, but also increased many adverse reactions. One of the possible reasons for the difference is that the baseline value of H-CRP of the patients included in the first two studies is high, while the latter does not limit the value of H-CRP, which results in the very low median value of H-CRP at the time of enrollment. This conjecture is also corresponding to the results of our study, that is, the higher the GPS score, the higher the MACEs incidence. Therefore, the selection of inflammatory indexes in anti-inflammatory treatment and the setting of cutoff value of intervention of these indexes are likely to be the focus of future research.

In addition to the inflammatory hypothesis based GPS, prognostic nutritional index (PNI) was also reported to have potential value in predicting adverse outcomes of pulmonary thromboembolism (*Hayıroğlu et al., 2018*). Therefore, in the future, combining the results of multiple prognosis scores may help patients with AMI to better avoid adverse outcomes.

There are some limitations to this study. First, it was a single-center, small-sample trial. Second, we only observed MACE and all-cause mortality during hospitalization. Long-term follow-up will provide a more comprehensive assessment of the predictive value of the GPSs. Third, because the cost of BNP testing is not covered by our country's medical insurance, it is mainly for the purpose of reducing the economic burden, so some patients do not test BNP. The main reason for the large number of missing EF value is that our ultrasound department can choose not to report the specific value of EF for patients with normal EF value, so we have to take measures to fill the missing data, but from the side analysis, if most of the missing EF values are in the normal range, then it is suggested that EF values should not be highly predictive of our end events. Finally, although the three GPSs were used in this study, the cutoff values of the CRP and albumin levels were derived from cancer patients, which may lead to their failure to achieve an optimal predictive value for AMI.

## CONCLUSIONS

Our study found that the GPS has a good predictive value for the occurrence of MACE during hospitalization in patients with AMI, regardless of STEMI or NSTEMI, the choice of PCI mode and the presence or absence of diseases that interfere with the GPS.

Therefore, the GPS may be used for risk stratification in the early stages of AMI. Large, multi-center and prospective studies still need to be performed to determine the ability of the GPS to assess prognosis in patients with AMI during hospitalization and follow-up.

## ACKNOWLEDGEMENTS

We thank MedSci for providing free sample size calculation tools for this study.

### Funding

This study was supported by the Science Technology Department of Zhejiang Province (2017C37130), and Hangzhou Health Science and Technology Project (2017Z10). The funders had no role in study design, data collection and analysis, decision to publish, or preparation of the manuscript.

### Grant Disclosures

The following grant information was disclosed by the authors:
Technology Department of Zhejiang Province: 2017C37130.
Hangzhou Health Science and Technology Project: 2017Z10.

### Competing Interests

The authors declare that they have no competing interests.

### Author Contributions

- Houyong Zhu performed the experiments, prepared figures and/or tables, authored or reviewed drafts of the paper, and approved the final draft.
- Zhaodong Li conceived and designed the experiments, analyzed the data, authored or reviewed drafts of the paper, and approved the final draft.
- Xiaoqun Xu performed the experiments, prepared figures and/or tables, authored or reviewed drafts of the paper, and approved the final draft.
- Xiaojiang Fang analyzed the data, prepared figures and/or tables, and approved the final draft.
- Tielong Chen conceived and designed the experiments, prepared figures and/or tables, authored or reviewed drafts of the paper, and approved the final draft.
- Jinyu Huang conceived and designed the experiments, prepared figures and/or tables, authored or reviewed drafts of the paper, and approved the final draft.

### Human Ethics

The following information was supplied relating to ethical approvals (i.e., approving body and any reference numbers):

The ethics committee of Hangzhou hospital of traditional Chinese medicine granted Ethical approval to carry out the study within its facilities (Ethical Application Ref: 2019KY028).
## Data Availability
The raw data is available in the Supplemental Files.

## Supplemental Information
Supplemental information for this article can be found online at http://dx.doi.org/10.7717/peerj.9068#supplemental-information.

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
