# Peer review of "Predictive value of three Inflammation-based Glasgow Prognostic Scores for major cardiovascular adverse events in patients with acute myocardial infarction during hospitalization: a retrospective study"

_PeerJ, doi:10.7717/peerj.9068_

## Round 0.1 · original submission · Major Revisions

· Academic Editor

Major Revisions

Dear authors,

In light of the reviewers' comments, I think your manuscript has high standards to be published in PeerJ, once some issues highlighted in their reports are correctly addressed. Consequently, my decision is MAJOR REVISION.

With respect and kind regards,
Dr Palazón-Bru (academic editor for PeerJ)

Reviewer 1 ·

Basic reporting

In this retrospective, single center cohort study, the authors aimed to investigate three kinds of GPS in patients who present with AMI. Results have revealed that the GPS has a good predictive value for the occurrence of MACE during hospitalization among these patients regardless of STEMI or NSETMI, the choice of PCI mode. The language of the article was sufficient. The tables and figures were also adequate. In terms of references, I can suggest that a recent article, which investigated the CAR (CRP/Albumin Ratio) in AMI and combines a similar parameters, should be added to the article. Hence, I recommend to cite below article in your text:
1-Prognostic efficacy of C-reactive protein/albumin ratio in ST elevation myocardial infarction. Scand Cardiovasc J. 2019 Apr;53(2):83-90. doi: 10.1080/14017431.2019.

Experimental design

In method section, I have some concerns;
1-Which method was used to measure CRP and albumin level?
2-In which time period was CRP and albumin level measured in the studied population?
3-Did you use the DeLong method to compare the AUC values of three kinds of GPS?
4- In order to show the model had an adequate calibration, the Hosmer-Lemeshow statistic should be presented.
5-How did you obtain the long-term MACE events in the study? With outpatient follow-up or telephone contact?

Validity of the findings

The conclusion was sufficient.

Additional comments

Thank you for this interesting study.
Some comments about the study
1- Did you exclude the patients who had an acute infection(s), auto-immune disease, and cancer?
2-You should add an information that multi-center and prospective studies are needed to determine the exact role of the GPS in patients with AMI.
3-You should add some clinical information to the discussion part. For example, patients with higher GPS might need more aggressive statin treatment or close-follow up after such event.

Reviewer 2 ·

Basic reporting

The study is well-designed and reported, despite several typo errors the language is fluent and convenient.
I have minor concerns about the study.
Another important index named as prognostic nutritional index (PNI) has been reported to have prognostic value even in patients with pulmonary thromboembolism.
PNI also includes albumin as a component, thus please add this score to discussion section with the help of a citation to 'A Novel Independent Survival Predictor in Pulmonary Embolism: Prognostic Nutritional Index.'.

Experimental design

The research is within the aims and scope of the journal.

Validity of the findings

The impact and novelty of the study is adequate.

·

Basic reporting

See below

Experimental design

See below

Validity of the findings

See below

Additional comments

The introduction is adequate for the research question of the paper.

The language is clear and understandable.

The structure is in accordance with the standards of the journal.

Line 48. In the Abstract, there is a mistake: No NSETMI…… it would be NSTEMI

In Materials and Methods, it would be necessary to clarify the characteristics of the GRACE score.

Line 120 to 121: in outcome event, “the primary endpoint event was MACE during hospitalization, which included cardiovascular death, all causes mortality. I think it can be a mistake…. All cause mortality is a secondary endpoint.

In References: there are references which the scientific journal´s names are in uppercase and others in lowercase

For the sample size calculation, in addition to indicate the web page where you calculated it, you must explain which kind of contrast you performed.

Selection of the predictors should not be based on the univariate analysis (see the PROBAST statement), therefore you must recalculate your results.

Missing data should be imputed, instead of carrying out a complete-case analysis.

Could you determine the outcome probabilities and the calibration of the models?

---

## Round 0.2 · accepted · Accept

· Academic Editor

Accept

All the points indicated in the previous letter has been correctly addressed. Consequently, your manuscript is accepted for publication in PeerJ. Congratulations!

Reviewer 1 ·

Basic reporting

I have no further comment

Experimental design

I have no further comment

Validity of the findings

I have no further comment

Additional comments

Thank you for revision
I have no further comment

Reviewer 2 ·

Basic reporting

The article is well-designed and presented. I want to thank the authors for their valuable contribution.

Experimental design

The research is well-designed, the methods are described with sufficient detail.

Validity of the findings

The underlying data have been provided well and the statistics is convenient.

Additional comments

After the revisions, the value of the manuscript significantly increased. The manuscript has a great value to the literature.